# Analysis of the Sustainability Level of Smallholder Oil Palm Agribusiness in Labuhanbatu Regency, North Sumatra

**Tennisya Febriyanti Suardi** [1,*] **, Lies Sulistyowati** [2], **Trisna Insan Noor** [2] **and Iwan Setiawan** [2]

1   Doctorate Program of Agricultural Science, Faculty of Agriculture, Padjadjaran University, Sumedang 45363, Indonesia
2   Department of Agro Socio-Economics, Faculty of Agriculture, Padjadjaran University, Sumedang 45363, Indonesia
*   Correspondence: tennisya12001@mail.unpad.ac.id

**Abstract:** Palm oil is a leading export commodity for Indonesia and the country's highest foreign exchange earner. This commodity also has a fairly important role in the economy in regions in Indonesia, especially in Labuhanbatu Regency as one of the centers of oil palm production in the Sumatran Archipelago. This study aimed to analyze the level of sustainability of smallholder oil palm agribusiness in Labuhanbatu Regency. The study was designed using quantitative methods and a proportionate stratified random sampling approach with 249 oil palm farmers as research respondents. The data were analyzed using the multi dimensional scaling (MDS) method assisted by the Rap—Palm Oil (RAPPO) software (version 1.0). The results showed that the multidimensional sustainability of the smallholder oil palm agribusiness in Labuhanbatu Regency was in the "fairly sustainable" category. This illustrates that the oil palm agribusiness needs stakeholder support to improve its sustainability status. The highest value is the economic dimension because it is the main source of livelihood for oil palm farmers, while the lowest value is the environmental dimension. The value obtained was influenced by the use of production inputs that were not in accordance with the recommendations.

**Keywords:** palm oil; sustainability; agribusiness; MDS

## 1. Introduction

To avoid compromising the ability of future generations to meet their own needs, sustainable development includes three things: improving the quality of life in a sustainable manner, using natural resources at low intensity, and leaving good natural resources for future generations. The main concern in sustainable development is the relationship between the economic, ecological, and social aspects [1].

The concept of sustainability is a multidimensional one that can be measured at various levels in space and time so that it can provide a wide field to look for measurement methods that address different aspects of sustainability [2]. The latest formulation at the global level related to sustainability is manifested in the concept of Sustainable Development Goals (SDGs), with 17 main goals that are a continuation of the Millennium Development Goals (MDGs) concept.

In order to support economic development by increasing the country's foreign exchange through export activities in the context of plantation crops, one of the plantation commodities that has an important role in the Indonesian economy is oil palm. Oil palm is one of the most widely developed plantation crops in Indonesia. Oil palm plantations also have many advantages. First, from oil palm plantations, palm oil will be produced, which is a raw material for various food and non-food products. Second, oil palm is the cheapest vegetable oil producer compared to other vegetable oil producing plants. Third, oil palm is very suitable to be planted in various regions in Indonesia. These advantages make oil palm a very important vegetable oil producer in the world oil industry.

There are three types of oil palm plantation concessions in Indonesia, namely people's plantations (PR), large state-owned plantations (PBN), and large private-owned plantations (PBS). Proportionally, 55.09% of the national oil palm plantations are managed by large private companies. Additionally, 40.62% are cultivated by smallholder plantations, and the remaining 4.29% of the national oil palm area is cultivated by large state-owned plantations [3].

Palm oil productivity in Indonesia tended to fluctuate from year to year during the 2014–2020 period, with an average growth rate of 0.37% per year. Large private and state plantations make the largest contribution to increasing oil palm productivity in Indonesia. Smallholder plantations have the lowest productivity compared to large private and state plantations, so there is a disparity in oil palm productivity between smallholder plantations and large plantations. In general, the development of oil palm area, production, and productivity affects the development of the oil palm industry by showing a very productive economic performance. The lucrative level of profit has encouraged very rapid growth of the oil palm plantation industry, causing the contribution of oil palm plantations to the economy to be quite good [4].

Oil palm plantations provide large revenues for the central government and local governments for forest areas allocated for oil palm plantation development, especially in Sumatra and Kalimantan. North Sumatra Province is one of the centers for producing palm oil in Indonesia. Historically, the first oil palm plantation in Indonesia was located in North Sumatra Province, precisely on the East Coast of Sumatra with a plantation area of 5123 hectares. Because of this, the Province of North Sumatra is included as a center for producing palm oil in Indonesia [5]. The development of the area of oil palm plantations in North Sumatra is in fact not supported by the productivity of the palm oil produced. Based on three types of oil palm land tenure, smallholder plantations have the lowest productivity compared to large state plantations and large private plantations in North Sumatra.

Farmers are upstream stakeholders of the oil palm agribusiness supply chain in smallholder plantations. Smallholder oil palm farmers choose how to use the land they own, the types of plants planted, and how to manage the plantation or land they own. Smallholder oil palm farmers are not bound by a contract with a particular mill or company, so they are eligible for assistance from the government. The management of oil palm plantations in North Sumatra also involves cities/districts that are continuously expanding their oil palm plantations. One of the districts in North Sumatra Province that is a center for palm oil production is Labuhanbatu Regency. Labuhanbatu Regency consists of nine sub-districts, all of which have oil palm plantations with a total area of 35,160 hectares. With 21,513 oil palm farmers, total palm oil production in Labuhanbatu Regency is 503,100 tons per year.

The productivity of smallholder oil palm plantations in Labuhanbatu Regency tends to fluctuate. It reached its lowest level of 14.3 tons/hectare in 2019. Fluctuations in oil palm productivity that occurred in Labuhanbatu Regency indicate the need for the development of an oil palm agribusiness system that is able to minimize fluctuations in oil palm productivity. Efforts to develop the oil palm agribusiness in Labuhanbatu Regency cannot be separated from various obstacles and challenges because they cannot be balanced with a change in the orientation of development from increasing income towards sustainable oil palm plantation practices.

The practices of oil palm plantations are aimed more at economic benefits to increase income, not balanced with environmental or social conservation aspects of the plantation sector, the low utilization of environmentally friendly technologies, and the lack of institutional roles in assisting smallholders in the development of sustainable oil palm plantations. Ideally, the balance of economic, social, environmental, technological, and institutional aspects or dimensions should be good and correlated [6].

As a major part of the sustainable development of oil palm plantations, in order to minimize negative issues and various problems that arise due to the ongoing development of oil palm commodities, smallholder oil palm farmers do not pay attention to the sustain-

ability aspects of the economic dimension, social dimension, environmental dimension, technological dimension and institutional dimension. For example, the use of non-certified seeds, the frequency of land ownership conflicts, the low use of environmentally friendly technology, lack of maintenance, crop errors, limited access to sales, as well as the lack of assistance and the role of related institutions that have an impact on the low yield of palm oil production obtained, are among the problems facing smallholder oil palm farmers [7].

In recent years, with the improvement in the quality of comprehensive human resources, various studies on the sustainability status of oil palm within the scope of smallholders have not yet been carried out through extensive research by various researchers. This research has shown that the level of sustainability of oil palm plantations is still included in the category of moderately sustainable and still needs substantial improvement from the various dimensions and indicators that influence it so that it can generate benefits for the farmer's economy by increasing income. Therefore, the design of sustainable oil palm plantation development for smallholders must be carried out to realize sustainable oil palm plantations from various dimensions [8].

This research was conducted to lay a useful theoretical foundation for research related to the sustainability of oil palm agribusiness by smallholders based on the theory of sustainable agricultural development and the theory of oil palm agribusiness. On the one hand, most research only focuses on the three dimensions of sustainable agricultural development by only explaining in general terms the level of sustainability [9]. For example, research on the sustainability of palm oil-based biodiesel only discusses in general the status of its sustainability level through three dimensions, namely economic, social, and environmental, but has not explained why only three dimensions are discussed, and what factors differentiate them.

Thus, this study aimed to analyze the level of sustainability of oil palm agribusiness through five dimensions of sustainable agricultural development by describing in detail the most sensitive attributes that affect the sustainability of oil palm agribusiness. These results provide an empirical reference to increase the level of sustainability of oil palm agribusiness from the development of sustainable oil palm plantations. The practical implications resulting from this research are as follows: (a) for business actors, this research is expected to be used as a basis for development in the application of cultivation aspects for farmers and increase understanding of smallholder oil palm agribusiness for other actors so as to increase production, productivity, and production quality, which ultimately will improve the welfare of farmers and the wider community as well as have implications for its sustainability; (b) for the government, this research is expected to be a policy consideration for the development of sustainable smallholder oil palm agribusiness by understanding the characteristics of farmers, the differences in agroecosystems, and considering the model that will be prepared as the basis for smallholder oil palm agribusiness development policies.

## 2. Materials and Methods

### 2.1. Sustainable Development

The concept of sustainability is applied to agricultural development mainly as a result of awareness arising from the negative impact of intensive farming systems on the environment and quality of life of rural communities. Sustainable agriculture is the management and utilization of agricultural ecosystems by maintaining biological diversity, productivity, regeneration capacity, vitality, and the ability to function so as to meet current and future needs of economic and social functions at the local, national and global levels and not endanger other ecosystems. Sustainability is also concerned with equality between generations. Currently, there are various choices of strategies to realize economic development in the agricultural sector; the development strategy that meets these characteristics is agribusiness development. Agribusiness development must be sustainable at least within three dimensions, namely the economic dimension, the social dimension, and the environmental dimension. With the changing conditions of the times, it is necessary to have a technological dimension and an institutional dimension in reviewing the assessment of sustainability

status. Thus, agribusiness development is not only for short-term interests, but also for long-term interests [10].

The development of sustainable agribusiness is based on five dimensions, namely the economic dimension, social dimension, environmental dimension, technological dimension, and institutional dimension: from the economic dimension to benefit and increase income, from the social dimension to integrating social knowledge through behavior change, from the environmental dimension to paying attention to the preservation of natural resources, from the technological dimension to using environmentally friendly technology, and from the institutional dimension to support and facilitate access to affordability [11].

The sustainability status of the oil palm agribusiness in this study was reviewed based on several theories and the results of previous studies, namely, which is seen through sustainability indicators consisting of economic dimensions, social dimensions, environmental dimensions, technological dimensions, and institutional dimensions. The economic dimension variables consist of indicators of productivity, income, employment, and access to sales of fresh fruit bunches. Social dimension variables consist of indicators of farmer regeneration, farmer empowerment, education level, and conflict resolution. The environmental dimension variables consist of indicators of environmental awareness, land use, infrastructure access, and flood frequency. The technological dimension variables consist of indicators of farmers' access to technology sources, price information systems, market information access, and fruit quality criteria. The institutional dimension variables consist of participation in farmer groups, participation in cooperatives, access to financial institutions, and accessibility of training and counseling [12].

### 2.2. Data

This study used a quantitative analysis design. The method used in this research was a survey. The data were obtained based on the results of a survey in the central provinces of oil palm plantations in Indonesia. This research was conducted from December 2021 to February 2022 based on nine central palm oil producing provinces. One province was selected, namely North Sumatra with one central area and two research locations. The central area was Labuhanbatu Regency. The two research locations were the rank sub-district and the northern Rantau sub-district.

The research sample was selected based on the difference in the amount of oil palm production obtained. The sampling technique used in the study was proportional stratified random sampling involving 249 oil palm farmers divided into two sub-districts and four villages.

### 2.3. Variable Selection

The variables studied in this study are described in detail from each dimension as shown in the following table (Table 1):

**Table 1.** Variable Description.

| Variable Dimension | Attribute | Score | Good | Bad | Description |
|---|---|---|---|---|---|
| Economy | Productivity | 1, 2, 3 | 3 | 1 | 1 = Low 2 = Medium 3 = Good |
| | Income | 1, 2, 3 | 3 | 1 | |
| | Employment | 1, 2, 3 | 3 | 1 | |
| | FFB Sales Access | 1, 2, 3 | 3 | 1 | |
| Social | Farmer regeneration | 1, 2, 3 | 3 | 1 | 1 = Low 2 = Medium 3 = Good |
| | Farmer empowerment | 1, 2, 3 | 3 | 1 | |
| | Level of education | 1, 2, 3 | 3 | 1 | |
| | Conflict resolution | 1, 2, 3 | 3 | 1 | |

**Table 1.** *Cont.*

| Variable Dimension | Attribute | Score | Good | Bad | Description |
|---|---|---|---|---|---|
| Environment | Environmental awareness | 1, 2, 3 | 3 | 1 | 1 = Low<br>2 = Medium<br>3 = Good |
| | Land use | 1, 2, 3 | 3 | 1 | |
| | Infrastructure access | 1, 2, 3 | 3 | 1 | |
| | Flood frequency | 1, 2, 3 | 3 | 1 | |
| Technology | Farmer access to technology resources | 1, 2, 3 | 3 | 1 | 1 = Low<br>2 = Medium<br>3 = Good |
| | Price information system | 1, 2, 3 | 3 | 1 | |
| | Access market information | 1, 2, 3 | 3 | 1 | |
| | Fruit quality criteria | 1, 2, 3 | 3 | 1 | |
| Institutional | Participation in farmer groups | 1, 2, 3 | 3 | 1 | 1 = Low<br>2 = Medium<br>3 = Good |
| | Participation of farmers in cooperatives | 1, 2, 3 | 3 | 1 | |
| | Access to financial institutions | 1, 2, 3 | 3 | 1 | |
| | Accessibility of training and counseling | 1, 2, 3 | 3 | 1 | |

### 2.4. Multi Dimensional Scaling (MDS)

In an effort to explain the research objective, namely sustainable smallholder palm oil agribusiness, apart from using descriptive analysis, it was analyzed using multi dimensional scaling (MDS) analysis developed by the Fisheries Center, University of British Columbia. The multi dimensional scaling (MDS) analysis aimed to analyze the sustainability of smallholder oil palm agribusiness in Labuhanbatu Regency. The economic dimension, social dimension, environmental dimension, technological dimension, and institutional dimension were based on indicators of sustainable palm oil agribusiness with the RAP— Palm Oil (RAPPO) ordinance technique modified from the Rapid Appraisal Technique for Fisheries (RAPFISH) program. The modification was carried out by developing or changing indicators in each dimension or dimension that would be used because it was adjusted to the system or topic and scope of research [13].

Sustainability analysis using the RAPPO (Figure 1) technique begins by reviewing, identifying and defining the indicators used. After that, an assessment (scoring) of the analyzed indicators is carried out. Scoring is based on the conditions that have been set in the RAPFISH technique. The scoring data are then processed using the software facility (software) RAPFISH (version 1.0), which is compressed (add-ins) on MS-Excel. According to the input of the indicator scores arranged in the Rap Scores matrix in the form of MS-Excel software worksheets, the data processing process takes place in the software.

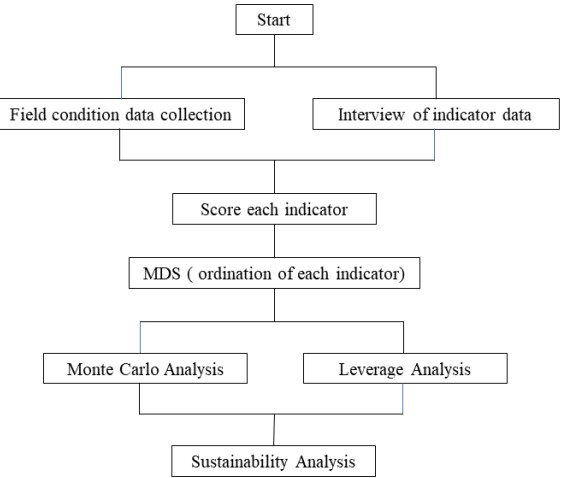

**Figure 1.** RAPPO Analysis Stage.

MDS analysis in RAPFISH gives stable results when compared to other multivariate analysis methods. In MDS, two points or the same object are mapped in single points that are close to each other. Conversely, objects or points that are not the same are depicted with points that are far apart. The ordination technique or distance determination in MDS is based on the Euclidean distance, which is in dimensional space, and the equation is as follows:

$$d = \sqrt{(|X_1 - X_2|^2 + |Y_1 - Y_2|^2 + |Z_1 - Z_2|^2 + \cdots)} \tag{1}$$

The ordinance of the object or point is then approximated by regressing the Euclidean direction ($d_{ij}$) from point $i$ to point $j$ with the origin ($\delta ij$). The equation is as follows:

$$d_{ij} = \propto + \beta \delta ij + \varepsilon \tag{2}$$

In order to regress the above equation, the least squared method is used alternately based on the roots of the Euclidean distance (square distance), called the ALSCAL method. This method optimizes the squared distance (squared distance = *dijklm*) to the squared data (origin point = *Oijklm*). In five dimensions (*ijklm*), it is called S-Stress according to the following equation:

$$S = \sqrt{\frac{1}{m} \sum_{k=1}^{m} \left[ \frac{\Sigma_t \, \Sigma_f \left( d_{ijklm}^2 - o_{ijklm}^2 \right)}{\Sigma_t \, \Sigma_f \, o_{ijklm}^4} \right]} \tag{3}$$

The squared distance is the Euclidean distance according to the equation:

$$d_{ijklm}^2 = \sum wka \left( x_{ia} - x_{ja} \right)^2 \tag{4}$$

After the ordination, the goodness of fit assessment is then carried out, namely the distance between the estimation point and the origin. The goodness of fit value reflects the magnitude of the S-Stress value of $R^2$. A good model is achieved if the S-Stress value is less than 0.5 (S < 0.25) and $R^2$ is close to 1 (100%). The stress value and the coefficient of determination ($R^2$) determine whether it is necessary to add variables to ensure that the variables used represent the properties of the object being compared.

The MDS analysis shows that the position of the sustainability point can be visualized through the horizontal and vertical axes described by the leverage analysis and Monte Carlo analysis. Leverage analysis was conducted to determine the sensitive indicators in influencing sustainability. Sensitivity analysis (leverage) in the MDS was carried out to determine the key indicators. The key indicators are obtained based on the results of leverage, as seen in the change in Root Mean Square (RMS) ordination on the X axis. The greater the change in RMS, the more sensitive the role of the indicator to the improvement of sustainability status.

Furthermore, the Monte Carlo analysis is carried out, namely the analysis of uncertainty. Monte Carlo analysis is an analysis used to estimate the effect of random errors in the analysis process carried out at a 95% confidence interval. In this case, Monte Carlo analysis is a simulation method to evaluate the impact of random error on all dimensions. This study used Monte Carlo analysis with the 'scatter plot' method, which shows the ordinance of each dimension.

Each indicator on each criterion is given a score based on the scientific judgment of the scorer. The score ranges from 1 (bad) to 3 (good) depending on the state of each indicator. The scores for each indicator are analyzed multidimensionally to determine one or several points that reflect the position of sustainability in the five dimensions studied relative to two reference points, namely the good (good) point and the bad (bad) point. Scores were analyzed by RAPPO to determine sustainability status below (Table 2).

**Table 2.** Criteria and Index Values and Sustainability Status.

| Index Value (%) | Status Category |
|---|---|
| 0–25 | Bad (unsustainable) |
| 25–50 | Less (less sustainable) |
| 50–75 | Enough (sufficiently sustainable) |
| 75–100 | Good (very sustainable) |

## 3. Results

The sustainability of smallholder oil palm agribusiness is an agenda that is able to describe the sustainability status based on predetermined sustainability indicators [14,15]. Assessment of the sustainability status of smallholder oil palm agribusiness in Labuhanbatu Regency was analyzed by the multidimensional RAP-Palm Oil (RAPPO) method and the five dimensions of sustainability, namely the economic dimension, social dimension, environmental dimension, technological dimension and institutional dimension. Statistical parameters in this study consisted of Monte Carlo analysis, the value of S-Stress and $R^2$. RAPPO analysis shows that the goodness of fit value reflects the magnitude of the S-Stress and $R^2$ values [16].

In a good model, the S-Stress value is <0.25 and $R^2$ is close to 1 (100%). The S-Stress value generated in each dimension and multidimensional has a value of <2.50, the smaller than 0.25, the better the S-Stress value. The values of S-Stress and $R^2$ indicate that all attributes used and analyzed dimensionally and multidimensionally have met the statistical criteria and are worthy of explaining the sustainability of smallholder oil palm agribusiness.

Table 3 shows that the S-Stress value is between 0.16–0.20, and the $R^2$ value is at 0.91–0.92, which means that the goodness of fit value in the RAPPO analysis has been fulfilled. The coefficient of determination ($R^2$) describes the ability of the attribute to explain and contribute to the sustainability of the system being analyzed, and if the S-Stress value is met, the attribute configuration can reflect the original data so that it can be stated that the indicators analyzed are accurate and can be accounted for statistically. The difference between MDS and Monte Carlo at the 95% confidence level or 5% error rate is between 1.03–2.36, so the impact of scoring errors in the analysis is relatively small. The value of the difference between the two analyses is <5%, so the results of the MDS analysis are adequate as an estimator of the sustainability index.

**Table 3.** Results (Goodness of fit) from RAPPO Analysis and Sustainability Status of Smallholder Oil Palm Agribusiness in Labuhanbatu Regency.

| | MDS | Monte Carlo | Difference | S-Stress | $R^2$ |
|---|---|---|---|---|---|
| Multidimensional | 59.61 | 58.58 | 1.03 | 0.20 | 0.91 |
| Economy | 70.61 | 69.23 | 1.38 | 0.16 | 0.92 |
| Social | 63.51 | 62.25 | 1.26 | 0.17 | 0.92 |
| Environment | 50.65 | 49.36 | 1.29 | 0.17 | 0.92 |
| Technology | 52.89 | 51.69 | 1.2 | 0.19 | 0.91 |
| Institutional | 52.21 | 49.85 | 2.36 | 0.18 | 0.92 |

*Status of Multidimensional Smallholder Palm Oil Agribusiness Sustainability*

The results of the multidimensional RAPPO analysis (Table 3) using the multi dimensional scaling (MDS) method resulted in a community oil palm agribusiness sustainability index value of 65.23; this value was in the range of 51–75 and included in the "fairly sustainable" category with an S-Stress value of 0.17, and the value of $R^2$ was 0.92.

Based on the results of the RAPPO analysis in Figure 2, the multidimensional value of the community oil palm agribusiness sustainability index was 59.61. This value is influenced by the calculation of the combined analysis of all dimensions (economic, social, environmental, institutional and technological), which is called multidimensional analysis

in the MDS analysis; values that are in the range of 50–75 are included in the "fairly sustainable" category.

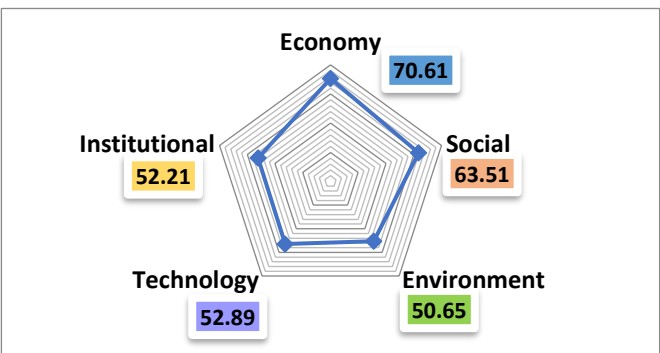

**Figure 2.** Index of Community Oil Palm Agribusiness Sustainability Index in Labuhanbatu Regency.

This is because of the five sustainability dimension variables tested in this study; the five sustainability dimension variables have a sustainability index value with moderately sustainable status, namely the economic dimension, social dimension, environmental dimension, technological dimension and institutional dimension. Thus, it can be concluded that smallholder oil palm agribusiness in Labuhanbatu Regency is quite sustainable [17].

Each dimension has attributes that are parameters for the sustainability of smallholder oil palm agribusiness in Labuhanbatu Regency. The value of the community oil palm agribusiness sustainability index was obtained based on an assessment of 20 sustainability attributes from each dimension.

The sustainability index value of each dimension projected in the flyover diagram (Figure 3) means that the further the sustainability point is away from 0, the greater the sustainability value. The fly chart is often referred to as a "radar" diagram where the closer the analysis distance to the zero point, the lower the sustainability and vice versa.

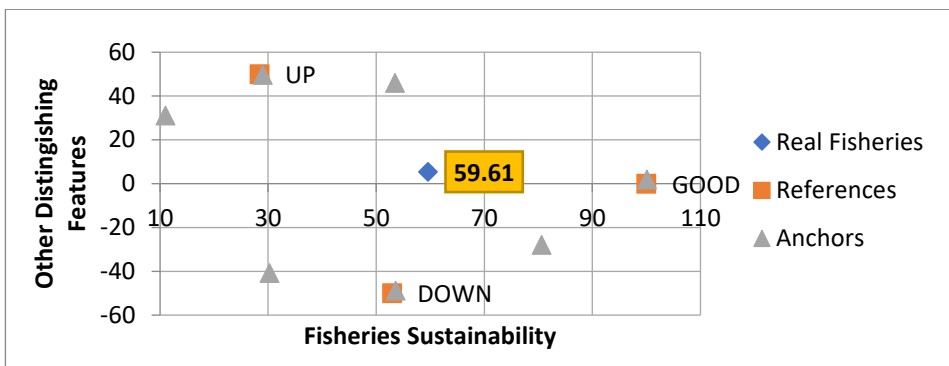

**Figure 3.** Flying Diagram of Community Oil Palm Agribusiness Sustainability Index in Labuhanbatu Regency.

Based on the flying diagram (Figure 3), it can be seen that the sustainability index value of the environmental dimension has the lowest value, followed by the institutional dimension, the technological dimension, the social dimension and the dimension with the highest value being the economic dimension. The value of the sustainability index of each dimension based on the flyover diagram is still not precise, meaning that each dimension of sustainability has not been applied evenly and in a balanced manner. The flyover diagram can describe the sustainability status of smallholder oil palm agribusiness in Labuhanbatu Regency in an integrated manner between various sustainability dimensions consisting of economic dimensions, social dimensions, environmental dimensions, institutional dimensions, and technological dimensions. Based on the results of the calculation of the value of the sustainability status of each dimension, it is described in the following figure (Figure 4):

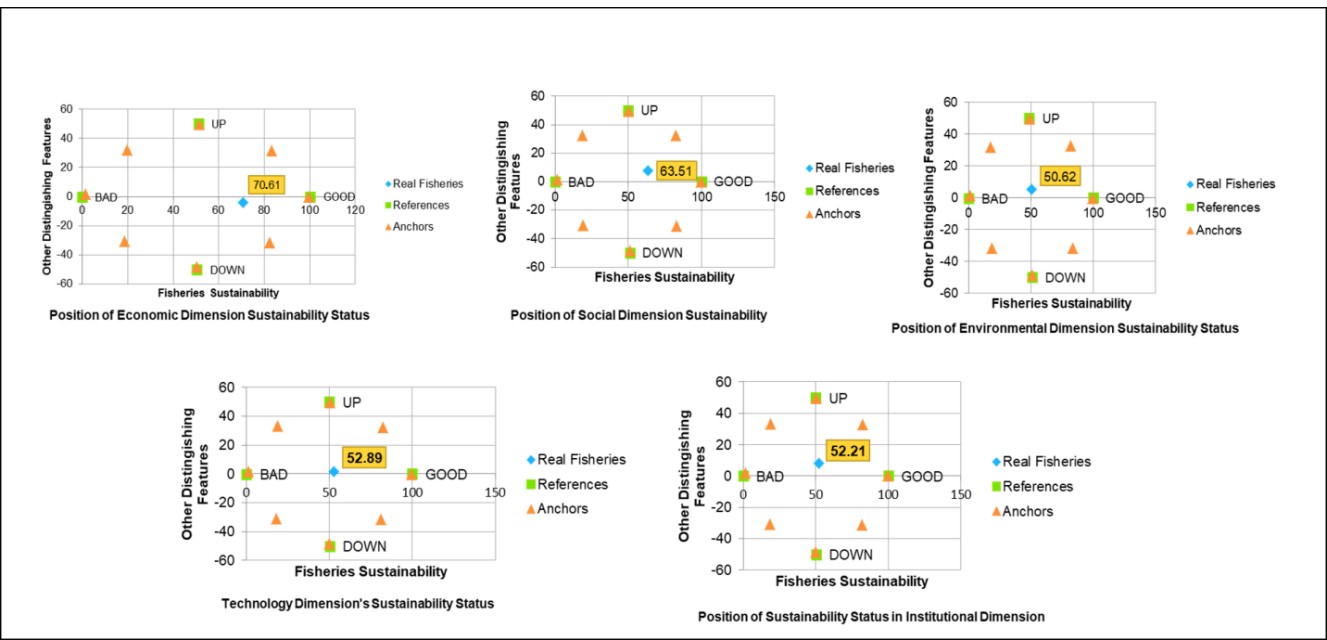

**Figure 4.** Position of Sustainability Status of Each Dimension.

The calculation of the most sensitive attribute values of the five dimensions of palm oil sustainability is shown below (Figure 5):

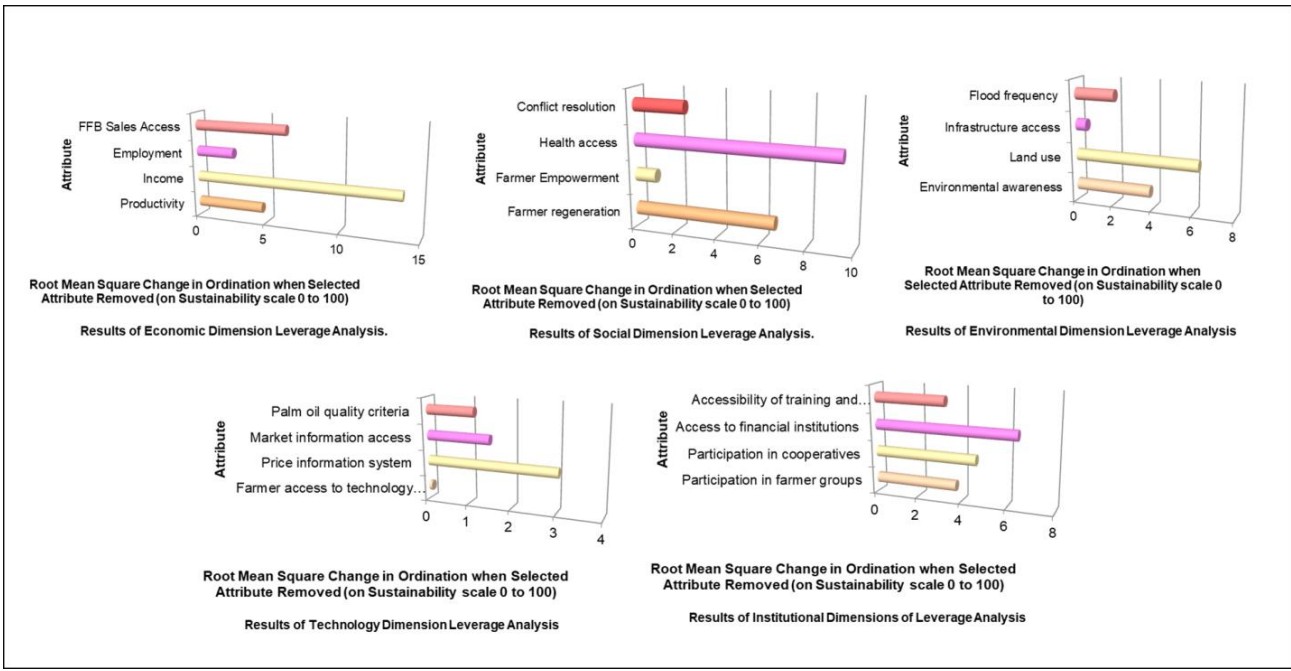

**Figure 5.** Results of Each Dimension Based on Leverage value.

## 4. Discussion

### 4.1. Economic Dimension

The economic dimension is one of the most important criteria in measuring the sustainability of smallholder oil palm agribusiness in Labuhanbatu Regency. The economic dimension is the ability of smallholder oil palm agribusiness to meet the needs of farmers in a sustainable manner. Smallholder oil palm plantations are said to be good or sustainable if they can improve the people's economic welfare [17]. The economic dimension is also one of the responsibilities in the concept of sustainable development, namely economic

success, which means the wise use of financial resources for the welfare of society [18]. There are four measurement attributes in the economic dimension that are analyzed by RAPPO analysis, including: (a) productivity, (b) income, (c) employment and (d) access to selling fresh fruit bunches [19].

The results of the analysis of the economic dimension of community oil palm agribusiness sustainability in Labuhanbatu Regency using RAPPO software on all attributes indicate that the value of the sustainability index of the economic dimension of smallholder oil palm agribusiness is 70.61 and is included in the "fairly sustainable" category, according to the range of values of 51–75.

Ordination analysis in the economic dimension with two iterations produces a value ($R^2$ = 0.9242), and the S-Stress value is 0.16 or 16%. Thus, the economic dimension analysis in this study shows the goodness of fit condition in the fair category. Analysis in multi dimensional scaling (MDS) is said to be good and acceptable if the S-Stress value is <25% or (<0.25) and the $R^2$ value is close to 1 or 100%. This shows that the attributes tested in the economic dimension can explain or approach the model 100% of the original model.

Sensitivity analysis of the economic dimension using the leverage analysis method in the RAPPO software shows that of the four attributes tested, it is known that there are two sensitive attributes that most influence the sustainability of smallholder oil palm agribusiness in Labuhanbatu Regency, namely income with an RMS value of 13.83 and FFB sales access with an RMS value of 6.39. The greater the value of the leverage analysis, the more sensitive the attribute in influencing sustainability. The previous description shows that in an effort to improve the sustainability status of the economic dimension of oil palm agribusiness, the community needs to pay attention to these two attributes [20].

The first attribute that most influences the sustainability of smallholder palm oil agribusiness in the economic dimension is income. Palm oil revenue is assessed based on whether it is the main source of income or there are other sources of income. On average, oil palm farmers in Labuhanbatu Regency make oil palm their main source of income and only depend on the sale of fresh fruit bunches (FFB) from farmers' plantations. Farmers who have other sources of income have higher incomes than farmers who only focus on oil palm plantations, so farmers tend not to have other sources of income outside the business [21].

The income of farmers from oil palm plantations ranges from IDR 3,800,000 to IDR 5,000,000 per month, and farmers own land ranging from 1–2.5 hectares, so it is only sufficient to meet household needs. However, if the income of oil palm plantations is not managed properly, it will threaten the sustainability of smallholder oil palm agribusiness from the economic dimension if it is not efficient in input preparation and rejuvenation.

The next attribute that most influences the sustainability of smallholder palm oil agribusiness in Labuhanbatu Regency on the economic dimension is access to FFB sales. All oil palm farmers sell FFB directly to middlemen (bandars) at a price of IDR 2700 per kilogram the difference in factory prices from middlemen is IDR 1900 to IDR 2300. This happens because oil palm farmers have cooperative institutions that are not active to oversee farmers in selling FFB. The role of cooperatives has a function in providing or distributing production facilities and marketing the products of FFB production. This phenomenon makes oil palm farmers in Labuhanbatu Regency only price takers.

### 4.2. Social Dimension

The social aspect is an important criterion in realizing sustainable smallholder palm oil agribusiness. The social aspect is one of the reference pillars of sustainable development that can contribute to rural development and poverty reduction [22]. The social aspect is part of the three main perceptions of sustainability, namely the social definition aimed at the continuous fulfillment of basic needs for security, justice, freedom, education, work and recreation. In this study, there are four attributes used to analyze the sustainability of the social dimensions of smallholder oil palm agribusiness in Labuhanbatu Regency, namely (1) farmer regeneration, (2) farmer empowerment, (3) education level and (4) conflict resolution [23,24].

Based on the results of the RAPPO analysis with four attributes on the social dimension, the value of the sustainability index of the social dimension of smallholder oil palm agribusiness in Labuhanbatu Regency is 63.51, including in the range of values of 51–75, so the social dimension category is quite sustainable.

The results of the RAPPO analysis test state that the position of the sustainability of the social dimension of smallholder oil palm agribusiness in Labuhanbatu Regency is in the range of 63.51. Ordination analysis, which is activated with a number of iterations twice, shows the goodness of fit condition in the fair category with a determination value ($R^2$ = 0.924) and the S-Stress value is 0.17 or 17%. These results have met the statistical rules in the analysis of multi dimensional scaling (MDS).

Based on the results of the sensitivity analysis (leverage) conducted on the four attributes of the social dimension, it is known that there are two attributes that have the highest leverage value. The attributes that are sensitive include the education level with an RMS value of 9.40 and the farmer regeneration attribute with an RMS value of 6.55. Sensitivity analysis (leverage) was conducted to determine the most sensitive attribute in influencing the sustainability of the social dimension of oil palm agribusiness. The greater the change in the RMS value, the more sensitive the role of these attributes will be to increasing sustainability status.

The most sensitive attribute influencing the sustainability status of the social dimension of smallholder oil palm agribusiness is the level of education. Education is an indicator of the quality of human resource development. The level of education usually affects the process of technology adaptation, that farmers' education is used to measure farmers' social conditions, which can indirectly influence the adoption of sustainability. The condition of smallholder oil palm farmers in Labuhanbatu Regency on the attribute of farmers' education level is included in the low category, which is dominated by elementary school graduates. The low level of education completed by oil palm farmers in Labuhanbatu Regency affects the absorption of adaptation and technological innovation related to oil palm plantations.

The next sensitive attribute is farmer regeneration. The condition of the regeneration of oil palm farmers in Labuhanbatu Regency has decreased significantly. The reason for this decline is the perception of farmers that they expect their children not to work on plantations like their parents did, the hope that their regenerated families can work in offices, as employees and others. The children of farmers are not willing to continue farming activities owned by their parents, so the children of farmers tend to sell the land inherited from their parents [25]. The description above also indicates that in terms of increasing the sustainability status of the social dimension of smallholder oil palm agribusiness in Labuhanbatu Regency, it is necessary to pay attention to the attributes of the level of education and regeneration of farmers. It is feared that this will have a negative impact on the sustainable management of smallholder palm oil agribusiness.

*4.3. Environmental Dimension*

One of the requirements in sustainable natural resource processing is to maintain the previous function of natural resources. In addition, it must have eco-efficiency criteria, which means efficient both economically and environmentally. Sustainable palm oil management must pay attention to the surrounding natural resources before exploitation, land clearing and management is underway. The environmental dimension is a key dimension because it can determine the balance of utilization of natural resources and environmental services. Environmental attributes were chosen to reflect how the use of natural resources and the environment has an environmental impact on sustainability.

The measurement of the environmental dimensions of community oil palm agribusiness in Labuhanbatu Regency uses four measurement attributes, which are analyzed using RAPPO analysis, including: (a) environmental awareness, (b) land use, (c) access to infrastructure, and (d) flood frequency [26,27].

Based on the results of the RAPPO analysis, the index value of the environmental dimension of smallholder oil palm agribusiness is 50.62 and is in the "fairly sustainable" category. This condition occurs due to the lack of understanding and concern of farmers on environmental sustainability that supports the sustainability of oil palm commodities.

Ordination analysis of the sustainability of the environmental dimension was carried out with a number of iterations twice, resulting in an $R^2$ value of 0.924 and an S-Stress value of 0.17 or 17%. Thus, the sustainability analysis of the environmental dimensions of smallholder oil palm agribusiness shows that the condition of goodness of fit is in the fair category and has met the requirements of a good multi dimensional scaling (MDS) analysis.

Leverage analysis is the next step after the RAPPO analysis. This analysis serves to identify the most sensitive attributes affecting the sustainability of the environmental dimension. The results of the leverage analysis state that the two most sensitive attributes in influencing the environmental sustainability of smallholder oil palm agribusiness are land use with an RMS value of 6.20 and environmental awareness attributes with an RMS value of 3.81.

The first attribute that affects the sustainability of the environmental dimension of smallholder oil palm agribusiness in Labuhanbatu Regency is land use. The area of agricultural land use is something that is very important in the production process or farming and agricultural businesses. Ownership or control of narrow land in farming is definitely less efficient than larger land. Land ownership or control is related to farming efficiency. The larger the area of land controlled, the more efficient the use of inputs or inputs will be. The condition of land tenure of oil palm farmers in Labuhanbatu Regency is such that land between ranges from 1 to 2.5 hectares, so it has an impact on the income of these farmers.

The next attribute influencing the sustainability of the environmental dimension is environmental awareness. Environmental awareness that is highlighted in this study is environmental pollution caused by spraying pesticides that are not in the right dose and not on time. It was found that some farmers had a concern that the use of high doses of pesticides would cause the grass or weeds to die quickly and be difficult to regrow and still spray pesticides during the rainy season, which is feared to pollute the river.

*4.4. Technology Dimension*

The sustainability of the technological dimension is the use and adoption of technology in smallholder oil palm agribusiness in facilitating oil palm farming [28]. Technology is an important requirement in achieving business efficiency in oil palm plantations. If the technological aspects are available and fulfilled, it will support the sustainability of oil palm farming. A good oil palm plantation must prioritize plantation cultivation technology in accordance with good agricultural practice guidelines. In order to meet the quality according to market demand, it is necessary to make efforts to increase productivity that is supported by the policy of implementing quality standardization of oil palm plantation management in order to obtain quality agricultural output or products so that the products produced can compete in the international market. This support is expected to provide high added value to farmers [29].

The measurement of the sustainability dimensions of smallholder oil palm agribusiness technology in Labuhanbatu Regency uses four measurement attributes, namely: (a) farmer access to technology sources, (b) price information system, (c) market information access, and (d) palm oil quality criteria [30].

The results of the RAPPO analysis on the sustainability of the technology dimension of smallholder oil palm agribusiness resulted in a technology dimension sustainability index of 52.89, which was included in the "fairly sustainable" category. This shows that the implementation of technology by oil palm farmers in Labuhanbatu Regency is still not optimal. RAPPO analysis on the technology dimension was carried out with two iterations, resulting in an $R^2$ value of 0.919 and an S-Stress value of 0.18 or 18%. Thus, the technology dimension analysis in this study shows the goodness of fit condition in the fair category.

Leverage analysis was conducted to determine the key attributes or the most sensitive attributes in influencing the sustainability of the technology dimension of smallholder oil palm agribusiness. Of the four attributes analyzed, the most sensitive attribute affecting the sustainability of the technology dimension was the price information system attribute with an RMS value of 3.03. This shows that in an effort to improve the sustainability status of the technology dimension, it is necessary to pay attention to these attributes.

A sensitive attribute that affects the sustainability of the technology dimension of smallholder oil palm agribusiness in Labuhanbatu Regency is the price information system. Smallholders generally do not have access to adequate information about technology, types and quality of inputs, input prices, and FFB prices, making it difficult for them to choose the most optimal combination of inputs to achieve maximum production [31]. Production inputs such as superior seeds and fertilizers are very difficult to obtain, and the price is quite high. The condition of oil palm price information in Labuhanbatu Regency only relies on price information from middlemen or dealers, so farmers can only sell their products to them. Access to price information is expected to enable smallholder oil palm farmers to sell their products to the appropriate sales institutions/places so that the farmers' incomes can be expected to increase.

### 4.5. Institutional Dimension

The institutional dimension is a part that must be linked in measuring the sustainability of smallholder oil palm agribusiness. The sustainability of the institutional dimension is the ability of group integration in indigenous palm oil agribusiness in carrying out institutional functions to facilitate plantation business activities [32,33]. Institutional parameters are indicators of the availability of legal and institutional instruments to encourage sustainable use of natural resources as well as the environment. When viewed from the institutional aspect, the availability of farmer associations, such as farmer cooperatives, short marketing networks, capital and financial institutions at the village level resulted in driving factors that could motivate farmers to increase the scale of production for a more modern economy with added value.

The institutional aspect is an important requirement for oil palm farming to be efficient so that it will be sustainable. Farmers need institutional roles to facilitate access in receiving assistance. The measurement of the sustainability of the institutional dimensions of smallholder oil palm agribusiness in Labuhanbatu Regency uses four measurement attributes, namely: (a) participation in farmer groups, (b) participation in cooperatives, (c) access to financial institutions, and (d) accessibility of training and extension workers [34,35].

Based on the results of the RAPPO analysis on the institutional dimension, it is known that the value of the sustainability index of the smallholder oil palm agribusiness institutional dimension of 52.21 is in the range of values of 51–75 and is included in the "fairly sustainable" category. RAPPO analysis in the institutional dimension was carried out with two iterations and resulted in an R2 value of 0.921 and S-Stress value of 0.18 or 18%. So, the goodness of fit value in the sustainability analysis of the institutional dimensions of smallholder oil palm agribusiness was in a fair condition and met the requirements of a good MDS analysis.

Sensitivity analysis was conducted to identify the most sensitive attributes in influencing the sustainability of the institutional dimensions of smallholder oil palm agribusiness. From the analysis of leverage, it is known that there are two attributes that are most sensitive to the sustainability of the institutional dimension, namely access to financial institutions and participation in cooperatives [36].

The first attribute that is most sensitive to the sustainability of the institutional dimension with an RMS value of 6.39 is access to financial institutions. This means that in order to improve the sustainability status of smallholder oil palm agribusiness institutions in the future, it is necessary to pay attention to the attributes of access to financial institutions in oil palm farming. The condition of oil palm farmers in Labuhanbatu Regency is still constrained regarding access to financial institutions to meet their farming capital needs.

These obstacles are caused by the remote location of financial institutions and the fact that most farmers borrow capital from relatives and moneylenders. There are three important aspects that need to be considered when studying smallholder access to finance [37]. First, identify the various financing schemes available to oil palm smallholders. Second, identify the perspective of lenders (formal and informal) regarding the effectiveness of different financing schemes in meeting the investment needs of smallholders, namely at the beginning of plantation establishment and operational costs of maintaining oil palm plantations. The third aspect is the identification of the behavior of smallholders when borrowing funds in relation to the cash flow of the oil palm plantations they cultivate [38,39].

The second most sensitive attribute affecting the sustainability of the institutional dimension is participation in cooperatives, with an RMS value of 4.61. The results in the field found that cooperative institutions in Labuhanbatu Regency, which are owned by smallholder farmers, are a combination of farmer groups (Gapoktan). The role of cooperatives is currently not active again even though it was previously active. This requires further review by stakeholders from the local government of Labuhanbatu Regency to improve and evaluate cooperatives in the oil palm plantation area. The mechanism for farmer partnerships through cooperatives with companies is in the form of farmers' obligations to sell oil palm FFB to companies through cooperatives with company standard quality. The company is obliged to buy FFB from cooperative members at a price according to government regulations.

## 5. Conclusions

The sustainability of smallholder oil palm agribusiness based on five dimensions of sustainable development, namely the economic dimension, social dimension, environmental dimension, technological dimension, and institutional dimension, was included in the "moderately sustainable" status. Multidimensionally, the economic dimension has the most sensitive influence on the sustainability of oil palm agribusiness through increasing farmers' income because oil palm farming is the main source of income for farmers to meet their daily needs. Furthermore, sequentially, the most sensitive multidimensional influence values after the economic dimension were the social dimension, the technological dimension, the institutional dimension, and the environmental dimension. In the social dimension, the influence of the education level attribute is most sensitive to the sustainability of oil palm agribusiness because the higher the level of education, the easier it is for farmers to innovate in their farming. In the technological dimension, the attribute of the price innovation system is the most sensitive to its effect on the sustainability of oil palm agribusiness because information on the selling price of fresh fruit bunches of oil palm must be structured and known periodically by farmers. Then, on the institutional dimension, the attribute that has the most sensitive influence is the role of financial institutions because smooth farming requires extensive capital and can be obtained from financial institutions that help farmers. Finally, on the environmental dimension, the most sensitive attribute is land use because it requires land suitability and land contours, and furthermore, on land use rights or leases that are legal and free from conflict. Therefore, to improve the sustainability of oil palm agribusiness, policy implementation models such as increased supervision of oil palm farming and increased product added value are needed from the side of farmers, government, and agribusiness actors as potential ways to improve sustainable agricultural development.

**Author Contributions:** Conceptualization, T.F.S. and T.I.N.; methodology, I.S.; software, T.F.S.; validation, L.S., T.I.N. and I.S.; formal analysis, T.F.S.; investigation, L.S.; resources, T.I.N.; data curation, T.F.S.; writing—original draft preparation, T.F.S.; writing—review and editing, T.F.S.; visualization, I.S.; supervision, L.S.; project administration, T.F.S., L.S., T.I.N. and I.S.; funding acquisition, T.F.S. All authors have read and agreed to the published version of the manuscript.

**Funding:** This research is self-funded, and APC is funded by Padjadjaran University.

**Institutional Review Board Statement:** Not applicable.

**Informed Consent Statement:** Not applicable.

**Data Availability Statement:** Not applicable.

**Conflicts of Interest:** The authors declare no conflict of interest.

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
