# Peer review of "Analysis of the Sustainability Level of Smallholder Oil Palm Agribusiness in Labuhanbatu Regency, North Sumatra"

_agriculture, doi:10.3390/agriculture12091469_

Round 1
Reviewer 1 Report
"Analysis of the Sustainability Level of Smallholder Oil Palm Agribusiness in Labuhanbatu Regency, North Sumatra" provides sufficient information regarding the underlying research, however, I suggest the following improvements:
1. Abstract seems incomplete, the level of writeup is too poor. Please improve the quality of communication. Please follow the Instructions for Authors. For convenience, the guide for the abstract is here"
"Abstract: The abstract should be a total of about 200 words maximum. The abstract should be a single paragraph and should follow the style of structured abstracts, but without headings:
1) Background: Place the question addressed in a broad context and highlight the purpose of the study;
-- Background of the study is not properly explained
2) Methods: Describe briefly the main methods or treatments applied. Include any relevant preregistration numbers, and species and strains of any animals used.
-- Methods are a little bit explained but not in the proper way. Line 9, "Research is designed with quantitative" is not suitable please elaborate in a scientific manner. The same is the case with Line 10, "The data analysis used is Multi Dimensional Scaling (MDS)"
3) Results: Summarize the article's main findings;
-- Main findings are not summarized.
and 4) Conclusion: Indicate the main conclusions or interpretations. The abstract should be an objective representation of the article: it must not contain results which are not presented and substantiated in the main text and should not exaggerate the main conclusions."
-- Abstract section does not include main conclusions and interpretations.
My suggestion is to rewrite this section completely.
2. Introduction
-- Line 19-20: "Sustainable development includes three things, including" repetition of includes/including is inappropriate. Moreover, the aspects indicated are not the things. Please reconsider the statement.
-- Line 22-23: "The main concern in sustainable development is the relationship between economic, ecological, and social" This is an incomplete statement. Economic, ecological, and social what? Please make it clear what exactly is being stated here.
-- Moreover, the introduction lacks a research question, the need for the study, and finally why is it important to study this issue? What are the main objectives of this research? What are the practical implications resulting from this study and what benefit will the farming community receive from the findings? and finally, how will this study add to the existing body of knowledge? All such information that increases the interest of the readers and proves its claim is missing.
I recommend authors add missing information and improve the quality of the write-up.
3. An important segment of a research article is the review of existing literature. This section is totally ignored by the authors. I suggest the authors add this piece of missing information.
4. Line 129-138: Please avoid using future tense as the work has already been done and now the authors are writing a report. The future tense is used in research proposals where authors are convincing their supervisors/funding agencies, about what they are going to do.
5. What is the rationale for employing multi-dimensional scaling analysis?
6. Please add numbering to the equations and also explain the equation in the text. For equations, please use the equation tool available in the MS Word.
7. Please explain more about Multi-Dimensional Scaling. Is it metric, non-metric, or generalized?
8. What is the reason for using Monte Carlo Analysis? Which of the research question has been addressed?
9. The results are nicely presented but still, there is a need to refine them. Moreover, the result section failed to provide a proper justification of the results.
10. Conclusion is very small and it should be stated in accordance with the findings of the study.
11. Policy suggestion/recommendation section is missing.
12. For the sake of guidance to future researchers, please add limitations of the study and advice for the researchers.
13. I suggest authors proofread and make write-up improvements.
Author Response
Dear our reviewer,
Dear our reviewer,
Thank you for being willing to review my manuscript very well. I realize that there are still many shortcomings that need to be fixed. Here I try to improve my manuscript in accordance with the suggestions you gave:
* Abstract: I have corrected referring to the consistency of the abstract in accordance with the provisions and amounting to almost 200 words
* Introduction: I have corrected it by referring to the core of the problem or research gap and led to the purpose of this journal paper
* Method: I have corrected it by referring to the specification of the data source and the selection of variables and tools used to process the data
* Result: I have corrected it by minimizing the image into a single unit in each dimension and discussing it in a multidimensional manner with a brief explanation
* Discussion: refers to discussions based on field studies combined with related theories so as to produce detailed writings, not only comparing with research only in related countries but also in other countries that implement sustainability
* Conclusion : Made concise according to the purpose. the conclusion discusses the level of sustainability included in the sufficient category and discusses the dimension that has the highest sensitivity value
So many revisions that I can explain, correct, and attach. Thank you very much for your time and willingness to review this manuscript. I really hope to receive suggestions for improvement from you for the progress of this manuscript.
Here's the manuscript I attached which has gone through the revision process
Thankyou so much,
Tennisya Febriyanti Suardi
Reviewer 2 Report
The issue raised by the authors is indeed an important one, and is interesting from both scientific and practical perspectives. Note that the authors have done a serious analysis. But the analysis itself will not be of interest to a wide audience, but is of a highly specialized nature.
There are the following comments and recommendations for the work.
The abstract should be an objective representation of the article. The abstract should introduce the reader to the research and hence state the methods used, some of the findings and some of the conclusions. It is not clear from the abstract which specific research methods the authors used. Which will not replace the mentioned Multi Dimensional Scaling. Increase the volume of the abstract to 150-200 words.
The authors should have read the publication's website more carefully with the requirements for abstracts.
The literature review refers to the central issue of the paper.
Unfortunately, the authors have not fully studied the theory of agribusiness sustainability from literary sources. Only 1 paragraph is allotted for this in the article (pp. 101-112). I would like to draw attention to the fact that the authors relied on research within their own country, without using the sources of authors from other countries on the topic under consideration. There are no views of various authors on this issue. Links to 4 sources is insufficient. Please for sources - 11 (line 100) and 12 (line 103) indicate the pagination. The review will be not of interest to other researchers studying sustainability.
The authors did not formulate their own position on what they mean by the sustainability of agribusiness. Thus, we do not see the contribution of the authors to the theory of this issue. There are duplicate references. Authors should have read the references requirements.
The use of scores to determine sustainability is questionable (line 182-184). The difference between the values ​​of indicators in the ranked distribution series is considered a constant value, estimated at one point, while in reality these differences can be purely nominal, especially at a high density of values.
These shortcomings can be avoided if, for example, a multivariate average or other statistical methods are used.
We can agree with the authors that not a point, but an interval estimate is expedient to characterize stability.
In the structure of the paper a Discussion section is absent, though it is rather significant. I believe that the section should be included and should contain the comparison of results in the context of previously published literature on the matter, as well as their discussion, which would allow authors to prove their points on the issue under consideration.
The conclusion should contain actual results. and should be connected with the paper aim. There should several important scientific results having novelty, future and application in the considered context. Overall this section requires an in-depth rethink and much further consideration on the results found by the research.
The article needs serious revision.
Author Response
Dear our reviewer,
Thank you for being willing to review my manuscript very well. I realize that there are still many shortcomings that need to be fixed. Here I try to improve my manuscript in accordance with the suggestions you gave:
* Abstract: I have corrected referring to the consistency of the abstract in accordance with the provisions and amounting to almost 200 words
* Introduction: I have corrected it by referring to the core of the problem or research gap and led to the purpose of this journal paper
* Method: I have corrected it by referring to the specification of the data source and the selection of variables and tools used to process the data
* Result: I have corrected it by minimizing the image into a single unit in each dimension and discussing it in a multidimensional manner with a brief explanation
* Discussion: refers to discussions based on field studies combined with related theories so as to produce detailed writings, not only comparing with research only in related countries but also in other countries that implement sustainability
* Conclusion : Made concise according to the purpose. the conclusion discusses the level of sustainability included in the sufficient category and discusses the dimension that has the highest sensitivity value
So many revisions that I can explain, correct, and attach. Thank you very much for your time and willingness to review this manuscript. I really hope to receive suggestions for improvement from you for the progress of this manuscript.
Here's the manuscript I attached which has gone through the revision process
Thankyou so much,
Tennisya Febriyanti Suardi
Round 2
Reviewer 1 Report
Thank you authors for incorporating the comments, but still a lot more is needed.
1. Abstract: Now this section has been improved by the authors in the light of suggestions and instructions for authors. Nice work.
2. My previous comment, "-- Line 19-20: "Sustainable development includes three things, including" repetition of includes/including is inappropriate. Moreover, the aspects indicated are not the things. Please reconsider the statement" has not been taken into account.
3. "-- Line 22-23: "The main concern in sustainable development is the relationship between economic, ecological, and social" This is an incomplete statement. Economic, ecological, and social what? Please make it clear what exactly is being stated here" has also not been taken into account.
4. "-- Moreover, the introduction lacks a research question, the need for the study, and finally why is it important to study this issue? What are the main objectives of this research? What are the practical implications resulting from this study and what benefit will the farming community receive from the findings? and finally, how will this study add to the existing body of knowledge? All such information increases the interest of the readers and proves its claim is missing" This comment is somehow addressed.
5. "An important segment of a research article is the review of existing literature. This section is totally ignored by the authors. I suggest the authors add this piece of missing information." The authors did not add a review of the literature.
6. This suggestion has also been ignored by the authors: "Line 129-138: Please avoid using future tense as the work has already been done and now the authors are writing a report. The future tense is used in research proposals where authors are convincing their supervisors/funding agencies, about what they are going to do"
7. Materials and Methods section has substantially been improved.
8. As per my previous suggestion, adding numbering to the equations and adding them in the text has not been taken into account.
9. The following comments also need to be incorporated:
-- The conclusion is very small and it should be stated in accordance with the findings of the study.
-- Policy suggestion/recommendation section is missing.
-- For the sake of guidance to future researchers, please add limitations of the study and advice for the researchers.
Author Response
Dear our reviewers,
Thank you very much for your time and willingness to review my manuscript. Here I attach the results of the second revision that I have corrected. I thank you for all suggestions for improvement and further input.
Thanks very much,
Tennisya Febroyanti Suardi
Reviewer 2 Report
The issue raised by the authors is indeed an important one, and is interesting from both scientific and practical perspectives. The authors conducted an in-depth analysis of the data on the stated topic I note that the authors have done some work to improve the manuscript. The authors finalized the abstract and included a discussion section. However, I still have some questions. The authors did not answer some of my comments presented earlier.
First, this paper briefly introduces agribusiness sustainability theory in the introduction section, though there are no literature reviews and findings from other studies to point out the objective and importance of this study. Any peer-reviewed journal paper must have a certain level of literature reviews to justify the contribution of this paper. The literature review refers to the central issue of the paper. Unfortunately, the authors have not fully studied the theory of agribusiness sustainability from literary sources. Thus, we do not see the contribution of the authors to the theory of this issue. The review will be not of interest to other researchers studying sustainability.
Second, there are duplicate references (line 1045-1050).
In the text, reference numbers should be placed in square brackets [ ].
Authors should have read the references requirements.
Third, the use of scores to determine sustainability is questionable (line 324-328). The difference between the values of indicators in the ranked distribution series is considered a constant value, estimated at one point, while in reality these differences can be purely nominal, especially at a high density of values.
These shortcomings can be avoided if, for example, a multivariate average or other statistical methods are used.
Nonetheless, We can agree with the authors that not a point, but an interval estimate is expedient to characterize stability. Although the authors do not provide justification for such a gradation.
Fourth, in conclusion, we should point out several important scientific results that are novel, promising, and applicable in the context under consideration. The paper also needs to include policy implications and recommendations for further studies in the conclusion section. In general, this section requires a deep rethinking and further consideration of the results obtained during the study.
The article needs serious revision.
Author Response

(The authors gave the same response as above.)
